# Network Approaches for Charting the Transcriptomic and Epigenetic Landscape of the Developmental Origins of Health and Disease

**DOI:** 10.3390/genes13050764

**Published:** 2022-04-26

**Authors:** Salvo Danilo Lombardo, Ivan Fernando Wangsaputra, Jörg Menche, Adam Stevens

**Affiliations:** 1Max Perutz Labs, Department of Structural and Computational Biology, University of Vienna, 1030 Vienna, Austria; salvo.lombardo@univie.ac.at; 2CeMM Research Center for Molecular Medicine of the Austrian Academy of Sciences, 1030 Vienna, Austria; 3Maternal and Fetal Health Research Group, Division of Developmental Biology and Medicine, Faculty of Biology, Medicine and Health, University of Manchester, Manchester M13 9WL, UK; ivan.wangsaputra@manchester.ac.uk; 4Faculty of Mathematics, University of Vienna, 1030 Vienna, Austria

**Keywords:** bioinformatics, networks, transcriptomics, epigenetics, integrative, development, DOHaD

## Abstract

The early developmental phase is of critical importance for human health and disease later in life. To decipher the molecular mechanisms at play, current biomedical research is increasingly relying on large quantities of diverse omics data. The integration and interpretation of the different datasets pose a critical challenge towards the holistic understanding of the complex biological processes that are involved in early development. In this review, we outline the major transcriptomic and epigenetic processes and the respective datasets that are most relevant for studying the periconceptional period. We cover both basic data processing and analysis steps, as well as more advanced data integration methods. A particular focus is given to network-based methods. Finally, we review the medical applications of such integrative analyses.

## 1. Introduction

In recent decades, a digital revolution has taken place in many scientific fields. In biology, we are able to produce large amounts of omics data (e.g., genomics, transcriptomics, proteomics, epigenomics, and metabolomics), which allow us to describe biological events in all their complexity. This has also led to a shift in the way we study, diagnose, and treat diseases in medicine, moving away from focusing solely on symptoms and clinical signs towards more holistic and data-driven approaches [1].

Such approaches are also indispensable to reveal the developmental origins of health and disease (DOHaD) (Figure 1A). Indeed, the earliest period of life is especially complex: It involves several closely interacting individuals, first and foremost the baby and the mother, but the father and the environment also play important roles and may condition different growth trajectories [2] (Figure 1B). For example, the phenomenon of imprinting in embryo development, i.e., the silencing of specific genes, leads to different outcomes depending on whether it happens on the mother’s or the father’s allele [3]. Several events, such as hypomethylation and microRNA regulation, can cause the loss of imprinting in the insulin-like growth factor 2 (IGF2) [4]. Normally, this gene is expressed only from the paternal copy and silenced on the maternal one. The expression of IGF2 on both alleles leads to widespread genomic, proteomic, and metabolomic changes responsible for various pathological conditions, such as hyperplasia, cancer, and embryo developmental disorders [5] (Figure 1C). The complexity of this example illustrates the need for experts from different fields to join forces and for an integrative view and analysis of the numerous and complementary layers of information that are at play [6].

The goal of this review is to facilitate the adoption of omics-based approaches in DOHaD research. We focus particularly on transcriptomics and epigenomics, given their widespread use and biological importance to DOHaD. We start with a brief introduction to transcriptomic data analysis. Next, we provide an overview of the plethora of epigenetic mechanisms at play in DOHaD. Finally, we introduce network-based methods as a tool for data integration from heterogeneous data sources and provide some concrete examples in the DOHaD context.

## 2. Transcriptomic View of Development and Analysis

Transcriptomics is the study of transcripts, or the expressed RNAs present in cells. From a transcriptomic perspective, human development can be seen as a continuous process in which fertilisation begins the transformation of the transcriptionally inactive oocyte into the active zygote, a process known as the maternal–zygotic transition [7,8]. This transition is but one of many windows of transcriptomic changes that the embryo undergoes as it develops and differentiates from a single cell to a complete organism [9,10]. These transition windows, often accompanied by epigenetic changes, have been suggested to be vulnerable periods in the embryo development, and disruptions during these periods can have lasting consequences in later life, as summarised in the DOHaD concept [11,12]. We describe how these transcripts (coding and non-coding RNA) regulate DNA expression and accompany epigenetic regulation mechanisms in the next section, while, in this section, we focus on the main principles of the bioinformatic pipeline used when facing a transcriptomic dataset and on the biological implications of these steps.

At its core, transcriptomic data analyses operate on the number of detected transcripts as a quantitative measure of gene expression. Historically, microarrays and RNA-seq have provided gene expression information for a group of cells. More recently, single-cell technologies allow us to measure transcript expression on a single-cell resolution [13,14]. The raw data of both bulk and single-cell technologies require several analytical steps, recapitulated in Figure 2 and described in the following sections.

### 2.1. Data Preprocessing

Several pre-processing steps need to be performed before transcriptomic data can be analysed. The first step, common to all platforms, is the alignment of transcripts to their source in the DNA, so that we may identify the genes that are being expressed. Over the years, many tools have been developed to perform this task [15]; widely used examples include STAR [16], BWA [17], and HISAT2 [18]. Alignments have to be performed against a reference database, typically an assembled genome and its annotations, for which commonly used sources are UCSC, NCBI, and Ensembl [19,20,21]. When attempting to replicate an analysis, care must be taken to use the same set of annotations as there are significant differences between them [22].

Once the transcripts have been aligned, the number of transcripts present for each gene can be tallied up to measure their expression levels. However, this number needs to be normalised before further analysis can take place to account for noise and biases of sequencing technologies. Noise may vary between batches due to variations in sample and preparation quality, and in some technologies, such as microarrays, high expression levels may cause saturation, resulting in a non-linear response. Controls, such as spike-ins or housekeeping genes, may be employed to provide a stable baseline for normalisation, though purely mathematical approaches, such as quantile normalisation, are also used [23].

Background noise is especially pronounced in single cell RNA-seq (scRNA-seq) datasets, as these platforms operate on a small volume per sample. The nature of this technology requires a normalisation step for comparing the counts of gene values across samples, such as reads per million/counts per million (RPM/CPM), which simply normalises the detected features to the total number of counts within the sample. Further development resulted in reads/fragments per kilobase million (RPKM/FPKM) [24] and transcripts per million (TPM) [25], which also take the gene length into account to avoid the overrepresentation of longer genes with more numerous exons. The Trimmed Mean of M-values (TMM) and Relative Log Expression (RLE) take this a step further, by making the assumption that the majority of genes are similar across samples and using them as control to allow for a more accurate inter-sample comparison [26,27].

### 2.2. Dimensionality Reduction

Transcriptomic datasets contain expression levels of thousands of transcripts and are thus by nature very high-dimensional. To enable more efficient downstream analyses, such as the identification of clusters, dimensionality reduction is often applied. The most commonly used dimensionality reduction methods are principal component analysis (PCA) [28], t-stochastic neighbour embedding (t-SNE) [29], and uniform manifold approximation and projection (UMAP) [30]. PCA is by far the simplest and fastest method and works by calculating the vectors on which the dataset is most variable. Though it is quick and preserves global distances, PCA is offset by losing information from the culled components and the linear modelling inherent in the data processing, which often results in a lower ability to separate different classes that may be contained in the data. It is also not very useful for visualising datasets in which the variance is spread over a large number of components, as most visualisations can only display two-to-three axes.

In contrast to PCA, t-SNE is both slower and does not preserve the global structure of the data. When operating on large datasets, t-SNE is often run as a second stage after other dimensionality reduction techniques, such as PCA, though the more recent Fourier-interpolated (FIt-SNE) implementation provides significant improvements in run time [31]. t-SNE is sensitive to changes in its own hyperparameters, which in turn must be selected carefully for each analysis to make sure the visualisation is fit for purpose [29]. Despite its shortcomings, t-SNE has a greater ability to create separation between clusters. This is especially beneficial for exploratory visualisations, though the fact that it does not preserve global distances means that the interpretation of visually observed clusters needs to be conducted carefully as differences in separation between clusters may not contain any significant meaning. Finally, t-SNE works best when used for visualisation only, reducing the dimension of the data to two or three at most, which means it is not ideal for downstream unsupervised clustering as it will only produce clusters that agree with its visualisation, potentially losing the underlying structure and relation between them due to the loss of global structure information [32].

Comparatively recent to the other two approaches is UMAP. As in the case of t-SNE, UMAP is a non-linear technique. However, in addition to creating strong local clusters, UMAP also attempts to preserve the global structure in the data and has lower computational time requirements [30]. In terms of single-cell analysis in particular, it is capable of preserving the continuity of cell subsets, which provides a more meaningful visualisation compared to t-SNE [33]. UMAP is thus often regarded as an improvement for most use cases, though t-SNE still shows greater local structure separation and the FIt-SNE implementation, in particular, has comparable, if not superior, speed.

Dimensionality reduced data are prime for visualisation, as with reduced complexity they can be more easily projected to a two-dimensional plane. This is a very useful feature, as a quick visualisation process can be a powerful quality control tool [32]. Due to the nonlinearity of the most commonly used dimensionality reduction methods, some caution needs to be exercised for interpreting the graphs produced. For example, a t-SNE based scatterplot produces exaggerated distances between local groups. From a quality control perspective, however, it can be very useful to rapidly assess whether or not the data are behaving as expected, i.e., are the clusters sensible, or whether or not the distribution is as expected from what is known about the dataset. Should there be a mismatch, it may point to other issues that need to be corrected first, for example, in the normalisation procedure.

### 2.3. Clustering

One of the most basic techniques for interpreting transcriptomic data is clustering, which identifies groups of similar points within the dataset. With the availability of a variety of different approaches for visualising and analysing transcriptomic data, care must be taken to select the technique most suitable for enabling insights relevant to the goal of the study. In transcriptomic data, there are two basic ways to perform clustering, either on the level of samples and or on the level of genes. Clustering samples is useful for identifying samples with similar gene expression levels, while gene clustering identifies genes with similar expression profiles across samples. Gene clustering is especially helpful in bulk sequencing experiments using older methods, such as microarrays, where sample clustering is often limited by low resolution. Sample clustering is especially suitable for single-cell datasets, for example, for identifying different cell types within a tissue. Two common clustering methods applied to clustering transcriptomic data are hierarchical clustering and k-means [34].

Hierarchical clustering is one of the most basic clustering methods available, where the data are structured into a dendrogram based on a statistical distance metric, such as Euclidean distance, and the clusters are decided by splitting the dendrogram at a certain depth. The dendrogram can be constructed by linking each data point together from the bottom up (agglomerative) or by splitting the complete dataset into smaller subgroups based on dissimilarity (divisive). While simple to understand, naive hierarchical clustering algorithms are computationally intensive and are not suitable for larger datasets, for which heuristics are often employed to speed up the process [35,36]. This technique was largely used in embryo developmental studies to investigate relationships between different embryological phases based on their transcriptomic similarity [37,38,39]. 

K-means clustering provides a faster algorithm compared to hierarchical clustering, in which the dataset is split into *k* cluster centres and assigns each data point to a cluster based on which cluster centre they are the closest to. Classical k-means clustering is an efficient technique when the number of clusters to be expected in the dataset is already known [40]. For this reason, it was applied for studying embryo development on a morphological level by imaging analysis [41], but also on a molecular level [42]. Despite these successes, the number of clusters of a dataset is not always known a priori and, to overcome this limitation, a variety of other methods exist [32], including network-based strategies that operate on graphs of transcriptomic data points constructed using K-nearest neighbour methods [43]. This approach revealed its potential especially in single-cell RNA-seq experiments [44], as it was able to identify known and new cell groups in different biological contexts, including embryo development [45] and artificial reproductive treatments [46].

While clustering is a very valuable tool for exploration, it is also often used as a stepping stone for downstream analysis. For example, annotating the clusters with genes that are highly expressed can lead to a better identification of what cell types are represented by the clusters using reference databases of marker genes. The list of expressed genes can itself be used in gene ontology (GO) enrichment analysis, which provides a way for looking up gene functions and can then be followed up with network analysis on the pathways and relations of the expressed genes, as well as overrepresentation analysis to find prevalent GO terms, which can lead to the identification of overexpressed pathways.

Clustering can also be performed on a gene level instead of a cell level, in which case genes are clustered based on the similarity of their sequences. This is especially useful for identifying the functions of novel genes as they may share similarities to known genes [47], but also for phylogenetic tree reconstruction [48]. It is also possible to group genes based on known biological processes instead, and then testing for the overrepresentation of certain processes within the sample.

### 2.4. Differential Expression Analysis

Differential expression analysis is a basic method for comparing gene expression levels between samples, and is able to produce useful insight regardless of resolution as long as there is clear delineation between samples [14]. By finding which genes are expressed at different levels between samples of different phenotypes, correlations between phenotype and genotype can be drawn. This idea of comparing the difference in gene expression levels also forms the foundation of much of the techniques explored in the following sections. This basic concept of comparing two (or more) groups has guided the latest biological discoveries in embryology. For example, some studies have tried to establish which genes guide a competent embryo implantation through in vitro fertilization techniques [49], while others have tried to correlate embryo degeneration with specific protein families, such as the heat shock proteins (HSPs) [50]. Other authors have tried to use well-known animal models used in embryology, such as Xenopus, to create an atlas of differential expressed genes during embryogenesis [51] and, finally, other studies have tried to translate these findings from animal models to human [52].

### 2.5. Trajectory Analysis

Trajectory analysis methods can be used to characterise datasets containing samples taken from organism(s) at different points in development [53]. While this is simple for experimental techniques that allow for multiple measurements on the same sample over time, it is impossible in single-cell experiments as the samples are destroyed upon sequencing. For the latter case, the concept of pseudotime is often employed, where the trajectory is constructed over various samples, instead of reflecting a real time series. Trajectory analysis is closely linked to the concept of a “developmental landscape”, in which cell populations roll down a landscape representing the entropy of their current state, heading towards their terminally differentiated fate [54]. As cells form a continuous lineage as they traverse through this landscape, the trajectory of cell development can be inferred by comparing the data taken from two different points. This comparison is typically performed by representing a cell’s gene expression profile as a vector, which allows for distances between them to be calculated. A range of approaches and packages are available for this purpose [55,56,57].

A recent technique referred to as RNA velocity analysis provides a novel approach to trajectory inference from a single snapshot in time. The technique utilises the ratio of unspliced to spliced messenger RNA (mRNA) to quantify the rate of change in gene expression and fill in the gaps between pseudotime slices [58,59]. Bridging these gaps can be very useful as some trajectory inference methods only work under the assumption that there is only a small difference between the different states [56]. Similarly to RNA velocity analysis, partition-based graph abstraction (PAGA) aims to detect trajectories from a single snapshot; however, it does this by connecting similar cells to each other in a network [60]. This allows to combine trajectory inference with other methods, such as clustering for the same data. Inferring transcriptomic trajectories in embryological datasets corresponds to the identification of developmental trajectories, which map cell state evolution over time. This is very important to try to understand biological events that occur in a very small window of time, such as the periconceptional period. Studies in mice have revealed the key genes whose transcriptomic changes lead to differentiation and organogenesis [45,61]. Similar studies in zebrafish have looked at a larger time window and contextualised the impact of some gene knockouts on cell fates [62].

### 2.6. Expressed Variation Analysis

Transcriptomic data can also aid in the identification of relevant genetic variations, in particular expressed single nucleotide variations (eSNVs) [63,64,65]. Compared to genome-wide association studies (GWAS) [66], these methods do not rely on having a large sample population. By operating at a much more granular single-cell resolution, they allow for detecting variations that are not present in significant numbers within a sample population. These traits enable the application of variant analysis on a smaller scale, such as separating cells from different individuals [67] and cancer identification [68,69]. When applied at the population level, it is also possible to define expression quantitative trait loci (eQTL), genetic variations that cause changes in expression levels [70], for example, using data collected in the Genotype-Tissue Expression (GTEx) project [71].

In conjunction with trajectory analysis, variation data can reveal further information. It has been shown that SNVs, both expressed and not, affect not only a gene’s transcription level, but its changes at different points in development, suggesting complex interactions with underlying regulatory mechanisms within the genome [72,73]. eSNVs are particularly important in this context as these expressed variants implicate phenotypic effects [74]. Similarly, it is also possible to compare the variations present in different stages of development to analyse the rate at which mutations are accrued at different stages, such as in studying the mosaicism present in human prenatal development [75].

## 3. Epigenetic View of Development and Analysis

The term epigenome refers to all modifications to DNA and histone proteins that modulate chromatin structure and genome function [76]. The epigenome thus represents a crucial nexus between genetic variation, environment, health, and disease. Indeed, chemical compounds, such as environmental exposures, can cause (ir)reversible changes in DNA structure, such as chromatin unfolding, which allows transcription factors (TFs) to bind their target leading to an increased transcription of the genes localised in the respective DNA region. Another example showing the importance of epigenetics is the variability of cell states within the same individual. While all cells in an organism share the same genetic information, they differ largely in terms of their expression and phenotypic manifestations. Epigenomic and transcriptomic data thus convey two different types of information, but can also be seen as two sides of the same coin. Integrating these two sides into a single framework remains an important and challenging task in the biomedical field [77,78,79].

Epigenetic mechanisms have gained growing attention from the scientific community in recent decades. In contrast to genetic mutations, epigenetic changes are plastic events that may occur multiple times during the lifetime of a cell and that can be reversible. These mechanisms are involved in numerous pathological processes [80,81,82] and therapeutic outcomes [83]. In the following, we review major biological processes that collectively constitute the epigenome (Figure 3).

### 3.1. DNA Methylation

DNA methylation is among the most studied and best understood epigenetic mechanisms. It contributes to gene expression regulation via an enzymatic reaction in which DNA methyltransferase (DNMT) catalyses the addition of a methyl group (-CH3) to a cytosine [84], causing DNA folding and the consequent inaccessibility by TFs, in turn silencing gene transcription in that particular region of DNA. In mammals, this process occurs only in dinucleotides CpG and is associated with gene inactivation [85,86]. Interestingly, each tissue seems to have a specific pattern of DNA methylation [87] that potentially changes over time, asserting the dynamic nature of this biological process [88,89] that can contribute to disease onset [90]. These dynamic methylation profile changes (epigenetic drift) contribute to cellular differentiation and tissue composition [91] and have been associated with development [92], ageing [93], and disease [94].

### 3.2. Non-Coding RNAs (ncRNAs)

The central dogma of molecular biology describes the flow of information from DNA to mRNA and finally to protein. Over the last decades, many additional mechanisms have been uncovered that can regulate and interfere with this linear process. Some of the main actors in this regulation are non-coding RNAs, which have been shown to regulate processes such as transcription, translation, and post-translational events [95,96]. There is a large variety of molecules that belong to this class and, surprisingly, some of them have been observed to be inheritable across generations [97,98]. They are often roughly classified based on their length into small (<200 nucleotides) and long (>200 nucleotides) non-coding RNAs. They can further be subdivided as follows:

siRNAs (small interfering RNAs) are very short sequences of RNA that are able to silence mRNA targets [99]. This phenomenon was first described in plants, fungi, and animals as RNA interference [100]. SiRNAs are generated from cutting long double-strand RNA (dsRNA), which can be generated from long hairpin RNA, genes, or pseudogenes. This process is executed by the biological machinery DICER.

snRNAs (small nuclear RNAs) are responsible for mRNA maturation [101] and participate in various fundamental biological mechanisms, such as splicing [102], TFs regulation [103], and the maintenance of telomeres [104].

SnoRNAs (small nucleolar RNA) are involved in chemical RNA modifications, such as methylation and pseudouridylation. The purpose and outcome of these modifications are still largely unknown, despite occurring in conserved regions of RNAs across species [105], as documented in recent databases that include information from seven different organisms [106], and integrate interaction information [107].

tsRNAs (transfer RNA derived small RNAs) are the most variable class of small non-coding RNA, having a repertoire of up to 150 modifications for each molecule [108]. Having been present since ancestral periods, they acquired different biological functions, ranging from bacterial development and viral infections to signalling molecules related to ageing, immunity, and disease [109]. Due to their cytoplasmic location and interaction with DICER, tsRNAs are regularly mis-annotated as miRNA [110], increasing the difficulties to fully understand their specific biological functions.

piRNAs (Piwi RNAs) are the largest class of small non-coding RNA molecules expressed in animal cells [111]. They are important to form protein complexes that silence transposons and other repetitive elements of the genome [112]. Estimates indicate that hundreds of thousands of different molecules belong to this class in mammals [113].

microRNAs (miRNAs) regulate gene expression by binding to mRNAs, thus suppressing their translation [114]. First described as early as 1993 [115], their basic mechanism of action has been known for some decades [116]. Still, many important questions remain unsolved [117], for example, how co-regulating miRNAs simultaneously regulate their target genes in different biological contexts.

Long non-coding RNAs (lncRNA) are RNAs with lengths exceeding 200 nucleotides and that are not translated into proteins. They have been implicated in many genomic processes, including parent-of-origin effects, alternative splicing, and tissue-specific gene expression [118,119]. In cancer, they have been observed to be particularly enriched for cis-expression quantitative trait loci (eQTL), which are often associated with genes regulating drug sensitivity [120]. LncRNAs are also associated with chromatin-modifying complexes [121] and histone methyltransferases [122].

### 3.3. Transposons

Transposons are regions of DNA that are repeated multiple times. They are also called “mobile elements” since they can change their position within the genome. They can be classified based on their mechanism of replication: Class I transposons, or retrotransposons, use a reverse transcriptase; and Class II transposons encode the protein transposase. Transposons play a major role in the diversification and evolution of the genome of different species, as well as individuals [123]. There are epigenetic mechanisms for avoiding unbalance in their transcription, for example, via methylation or zinc protein regulation [124]. However, despite their unpredictable jumps across the genome that may interrupt gene sequence and cause shifting mutations, certain diseases have shown specific associations with transposons, such as haemophilia [125] and Alzheimer’s disease [126].

### 3.4. Chromatin Modifications

With an estimated length of around 3 m [127], the DNA must be carefully folded to fit inside a cell. Together with histone proteins which aid in this process [128], it forms the chromatin complex. Chromatin is an extremely dynamic entity, whose changes lead to open or closed regions of the DNA, which in turn directly affects gene transcription. Chromatin modification is a set of epigenetic processes that govern many aspects of DNA replication, transcription, and repair. In eukaryotes, the basic unit of chromatin, the nucleosome, is comprised of 147 bp of DNA wrapped around a histone octamer made of two dimers of H2A and H2B and a tetramer of H3 and H4 proteins [129]. The interaction between DNA and histones occurs at the amino-terminal (N-terminal) tail of histone proteins and, for this reason, chemical modifications here, such as (de-) acetylation, phosphorylation and methylation, would change chromatin conformation, influencing various biological processes [130]. It is known that histone modifications are related to inheritance from mother to daughter cell and that this is influenced also by the environment, but the exact steps how this phenomenon occurs is still to be understood. These processes are crucial in development [131] and disease onset [132], but systematic large-scale studies remain scarce [133,134].

## 4. Network Models of the Epigenome

In light of the diversity of the biological mechanisms outlined above, it is clear that no isolated process or particular dataset alone will be able to provide a comprehensive picture of the developmental origins of health and disease. Indeed, after decades of biological research following a reductionist paradigm, a more holistic, systems-based framework is required [135]. Network theory can provide such a framework [136]. Networks are a general mathematical formalism for representing relationships (links) between objects (nodes). Important examples in biology and medicine range from protein–protein interaction (PPI) networks representing physical interactions between proteins [137] or gene regulatory networks representing transcription factors binding to DNA [138], to signalling networks of immune cells [139] or networks of organs linked by metabolism [140]. More generally speaking, we can distinguish between physical networks, where the links represent a direct physical relationship (e.g., protein interactions) and functional networks, where links represent indirect relationships (e.g., co-expression networks) [141] (Figure 4A).

The abstraction of the complex biological machinery in terms of networks allows us to systematically investigate both global and local connection patterns and their respective biological implications [142]. For example, highly connected nodes (hubs) in PPI networks typically correspond to proteins with multiple biological functions. These proteins have been shown to be more likely essential [143,144], so that network analyses can help to identify the crucial genes involved in particular biological mechanisms [145]. Similarly, groups of densely interconnected nodes (network communities) and recurrent structures in different parts of the network (network motifs) have been shown to correspond to genes participating in the same biological process and allowed for the identification of fundamental building blocks of biological pathways, respectively [146,147,148] (Figure 4B). Network communities can also aid in the identification of genes that are involved in a particular disease and form a so-called disease module [149,150,151].

### 4.1. Gene Regulatory Networks (GRNs)

We can observe from the above that the relationship between genome and proteome is not a simple linear process, but that many factors and feedback mechanisms are involved. These can be external factors, internal molecules of the organism, and importantly also interactions among the genes themselves [152]. From a theoretical point of view, we can define the gene regulatory network as the wiring diagram that controls the collective gene expression [153]. In the early 1970s, Kauffmann and colleagues provided a first theory for GRNs [154]. Specifically, they considered Boolean networks and showed that a complex collective behaviour can emerge from simple logical operators among the individual components. The introduction of high-throughput technologies enabled the combination of theory and large-scale data [155,156,157]. In the last decade, various methods have been proposed to identify the (generally non-linear) functions that govern gene expression [158]. Today, we can incorporate a plethora of different types of omics data, such as RNA-seq, ChIP-seq (chromatin immunoprecipitation sequencing for identifying DNA binding sites), or ATAC-seq (assay for transposase accessible chromatin sequencing to identify open chromatin regions). This led to a somewhat broader definition of gene regulatory networks that includes various biological mechanisms that influence gene expression [159], such as transcriptional regulatory networks [160], protein interactions [161], microRNA networks [162], and metabolic networks [163].

GRNs can be used to better understand the molecular machinery governing cell states and to guide new screening experiments [164]. They may identify subnetworks and pathological pathways that can help to identify network-based biomarkers [165]. Gene regulatory networks also play an important role in development and ageing. In addition to dynamic changes over time, interindividual variation also needs to be considered. Methods that account for this include LIONESS (Linear Interpolation to Obtain Network Estimates for Single Samples), an approach able to distinguish the individual variability within a group [166]. This tool is part of a group of algorithms for GRN analysis called netZoo package [167] that also provides functionalities for investigating tissue specificity or multi-omic data integration.

### 4.2. Network Approaches for Interpreting DNA Methylation Profiles

Dynamic changes in methylation profiles (epigenetic drift) have been mapped on PPI networks, showing that mainly peripheral genes with low connectivity values are affected that fall into a number of connected network communities [168]. This enabled the identification of age-associated hot spots in stem-cell differentiation pathways [168]. Despite these successes, the mechanisms by which different methylated CpG regions influence remain poorly understood. To address this, the concept of Functional Epigenetic Modules (FEM) was proposed to identify gene modules of coordinated differential methylation and differential expression in the context of the human interactome [169].

To better understand how methylated genes are influencing each other, correlations between the demethylation status of a pair of genes can be considered. Computing all possible DNA methylation status comparisons leads to the so-called Co-occurrence and Mutual Exclusivity (COME), a table specifying for each gene pair whether their methylation status co-occurred or is mutually exclusive. This procedure was recently applied in cancer research using 14 main cancer types from The Cancer Genome Atlas (TCGA) [170], revealing a new way to stratify patients distinguishing several classes with distinct epigenetic trademarks that correspond to distinct clinical outcomes [171].

Similarly, DNA methylation correlation profiles were used to build a co-expression network (DNA methylation interaction network) in ovarian cancer, breast cancer, and glioblastoma multiforme, predicting new common prognostic genes [172]. Recently, the integration of differentially methylated genes and differentially expressed genes was used to identify new biomarkers in leukaemia [173]. Another approach for methylation and expression profile integration consisted of a multi-layer network approach called “Epigenetic Module based on Differential Networks (EMDN)”. The method first builds separate co-expression co-methylation networks. Then, the modules of densely connected node groups that are shared between the two are identified. The results indicate the potential of this procedure for finding new disease-associated genes in breast cancer [174].

Another challenge in the area of DNA methylation is to integrate large-scale population studies [175,176]. Similar to GWAS, which have been extensively used to find new genomic variants associated with specific phenotypes [177], epigenome-wide association studies (EWAS) have also been proposed to help to discover new aberrantly methylated genes [178]. In this context, a network-based algorithm (NEpiC) was proposed for combining methylation profiles from EWAS and PPI modularity. For each gene, the algorithm computes a score to identify differentially methylated genes that are then mapped on the PPI. Then, the modularity of these genes is evaluated and a prioritisation algorithm based on the connectivity is applied [179].

To encourage the usage of these and other tools and discoveries in a clinical setting and by people without highly specialised bioinformatics training, a number of user-friendly platforms have been recently developed. A prominent example is DNMIVD [180], an interactive DNA methylation visualisation resource providing information regarding DNA methylation-based diagnostic and prognostic models based on different cancer types from TCGA, expression-methylation quantitative trait loci (emQTL), pathway activity-methylation quantitative trait loci (pathway-meQTL), differentially variable and differentially methylated CpGs, survival analysis, and FEMs (from PPI and COMEs) [181].

### 4.3. Modelling Non-Coding RNA Interactions

As introduced above, many epigenetic molecules belong to the wide category of non-coding RNAs. For many non-coding RNAs, little is known about their biological activity or interactions with other biomolecules. The most information is available for miRNAs, with over 2000 miRNAs discovered in humans, many of which are associated with diseases [182,183]. Accordingly, miRNAs have also been the focus of network-based studies, although the general methodologies are likely to be applicable to all classes of non-coding RNAs.

An important challenge is to identify elements that jointly regulate their target genes in different biological contexts. For miRNAs, a solution has been proposed that starts from creating a network in which two miRNAs are connected when they share a significant number of gene targets, as determined by sequence complementarity and co-expression patterns [184]. The Molecular COmplex DEtection algorithm (MCODE) was then used to identify 12 different miRNA modules in this network. The cooperativity of miRNAs within a module was evaluated by their shared TFs and the functional similarity of their target genes. Similarly, synergistic miRNA-miRNA networks have been proposed, in which connections are based on common targets with similar biological functions and close proximity in the PPI network [185]. It was shown that disease associated miRNAs are located at central positions in the resulting network and that miRNAs associated with the same disease tend to form connected clusters [186].

Network science also provides an arsenal of tools for finding new miRNA–diseases associations [187]. For example, one can construct a bipartite network in which miRNAs are connected to diseases based on their reported associations. New miRNA–disease associations can then be predicted using a combined score that takes functional similarity among miRNAs into account, as well as similarity among diseases [188]. These predictions can be further improved by including network topology features, such as neighbour similarities and network distance [189]. In this fashion, new functional annotations and similarities between miRNA pairs were discovered in the context of prostate cancer [190].

Another important class of non-coding RNAs are lncRNAs. One of the first attempts to study the biological functions of lncRNA in a large-scale fashion was presented in [191]. In that study, the authors built a coding–non-coding gene (CNC) co-expression network for predicting the biological role of 340 lncRNAs. The function of a particular lncRNA was inferred based on patterns of co-expression and genomic co-location, using the gene ontology annotations of the coding genes in its immediate neighbourhood in the CNC network. The concept of predicting biological functions based on the interactions between lncRNAs and proteins has further been applied using a random walk with a restart algorithm on a lncRNA–protein interaction network [192].

Despite many promising initial results, the predictive power of these methods remains limited. One reason for this is likely that they rely on a direct link between non-coding elements and coding genes, for which data are scarce. This can be mitigated by considering additional datasets, for example, phenotypic similarities [193]. Similarly, the focus on a single class of biomolecules is often a limitation. Indeed, many lncRNAs have been shown to interact with other non-coding elements, such as miRNA [194]. An approach using a combined lncRNA–miRNA–mRNA interaction network found that predictions based on all three data performed better than using only lncRNA–protein interactions and can be used to identify clinical biomarkers in the context of breast cancer [195]. Along the same lines, the integration of pre- and post-transcriptional information into a lncRNA functional similarity score was used to predict disease associations [196]. These approaches were later expanded to include both human and mouse data [197,198].

### 4.4. Network Approaches for Chromatin Modifications and Transposons

To date, more than a hundred enzyme complexes, grouped in at least eight different classes, are known to catalyse enzymatic reactions in histones and cause changes in the DNA structure [199]. A collection of manually curated genes–diseases associations related to chromatin modifications is available from [200,201]. These resources can also be used to construct networks for elucidating the relationship between chromatin modifications and disease [202]. For example, the conformational state of chromatin was shown to be responsible for the switching to an inflammatory phenotype in macrophages and that the underlying mechanism that regulates this process is governed by a transcriptional regulator network [203]. While only a few network studies have been performed in this area of epigenetic regulation, the growing amount of related genes–disease annotation opens up new doors for systematic investigations [204]. The potential of investigating genetic mobile elements was touched only on the surface and it could unravel many open biological questions. For example, Levy et al. have developed a computational framework that is able to identify retrotransposons playing a key role in species evolution [205].

## 5. Towards an Integrative Analysis across Biological Hierarchies

Several of the examples above showed the potential of combining different data sources and corresponding biological mechanisms. The ultimate goal, of course, is to integrate all relevant layers of biological information into a single comprehensive picture [206]. Indeed, no single dataset can capture the complex and high-dimensional nature of the biological processes involved in health and disease [207]. The integration of data from various sources poses both technical and conceptual challenges [208]. In the following, we highlight recent developments in this area, with a particular focus on examples that involve the various mechanisms introduced above.

From a data science perspective, data integration can often be boiled down to finding correlations and corresponding trends between different datasets. A basic way to achieve this is through a particular type of matrix operations, so called matrix factorisation, which provides a dimensionality-reduced model of the relation between features in different data. A widely used method for this is joint non-negative matrix factorisation. While simple to implement, being based only on matrix multiplication, it requires large amounts of computing resources and proper care needs to be taken during normalisation. More advanced variations of this technique that pose less restrictions on the data that can be used include the iCluster package [209].

There are also network-based methods for data integration. These methods build on the fact that many biological processes have a direct network representation, but also on the interpretation of networks as a visual and abstract representation of data matrices, allowing an intuitive, but also formal, approach to biological data analysis. For example, MpDisNet is a network-based methodology for identifying disease–disease relationships by integrating four different biological networks: disease–miRNA network, miRNA–gene network, disease–gene network, and the human interactome [210]. The methodology revealed that comorbidities for pulmonary diseases were driven by miRNA-mediated pathobiological pathways. It thus identified a new type of disease–disease relationship that filled a gap between genome and phenotype using miRNA data as a bridge.

Another approach for integrating different types of biomolecules uses cooperative networks, in which the different components contribute to a common goal. This concept was applied in cancer research to elucidate the mechanisms responsible for the upregulation of oncogenes [211]. Here, different components contribute to the cancerogenic phenotype: chromatin opening, the recognition of the gene-specific DNA motif, the creation of a scaffold between histones, and the constitution of the transcriptional complex. The method involves several types of enrichment analysis to obtain a prioritisation of the key elements that regulate each epigenetic step of transcription regulation. For example, motif enrichment was used to determine which transcription factor motifs are significant for certain promoters, while the transcription factor target analysis defined the transcription factors that govern a set of target genes.

Network-based data integration has been successfully applied not only in oncology, but also in other medical fields. In virology, for example, genetic–epigenetic interactions were used in a cross-species integration between Epstein–Barr virus and human cells [212]. The resulting network was obtained by merging PPIs, gene–miRNA interactions, gene–lncRNA interactions, and host–virus cross talk networks. This allows for the identification of dynamic epigenetic patterns, suggesting an initial epigenetic inhibition by viral proteins that results in an immune response dysregulation of the host. The analysis further uncovered the most active viral proteins and miRNAs responsible for resistance mechanisms against the host’s defence. In cardiology, analogous methods were used to distinguish different heart failure phenotypes, proposing the “EPi-transgeneratIonal network mOdeling for STratificatiOn of heaRt Morbidity” (EPIKO-STORM) [80].

While this review mainly focuses on network approaches to integrative analysis, other methods for data integration are also of interest. An approach that is currently undergoing rapid development is the use of machine learning. This computational approach is based on the capability of the algorithm to identify patterns from a part of the dataset (training set) that is used to learn fundamental characteristics of the shape of the data that can be used for classification and integration tasks [213]. Such an approach is able to integrate data beyond transcriptomic and epigenomic data, having been applied to other omic data types, such as genomic, proteomic, and metabolomic data [214,215]. As such, machine learning presents itself as a promising approach to integration, though having the limitation that the training dataset should be large enough to contain all the representative features of the real dataset, which is not always the case [216].

In conclusion, using these integrative approaches in a plethora of transcriptomic (Table 1) and epigenetic data (Table 2) can help to understand the pathophysiological mechanisms behind common conditions of the periconceptional period and pregnancy, such as placenta percreta [217], pre-eclampsia [218], and the pregnancy-induced hypertension syndrome [219].

## 6. Conclusions

Multi-omics data are becoming more and more prevalent. Even nowadays, there is a massive amount of transcriptomic data as well as a growing amount of diverse epigenetics data publicly available to the community. These data represent a treasure trove for the community, aiming to unravel the developmental origins of health and disease. Mining these data requires approaches for their systematic integration and interpretation. Network-based methods are particularly promising for these purposes. We hope that the examples highlighted in this review may serve as an inspiration and motivation for this exciting area of research, showing that integrative analysis enables insights that are not accessible from any one dataset or biological process alone. In the future, the scale and diversity of relevant dataset will only grow, enabling, for example, to connect molecular data with epidemiological findings. The insights that could be gleaned from such datasets have potential to impact large scale populations, and to help in the prevention, prediction of disease onset and intervention.

## Figures and Tables

**Figure 1 genes-13-00764-f001:**
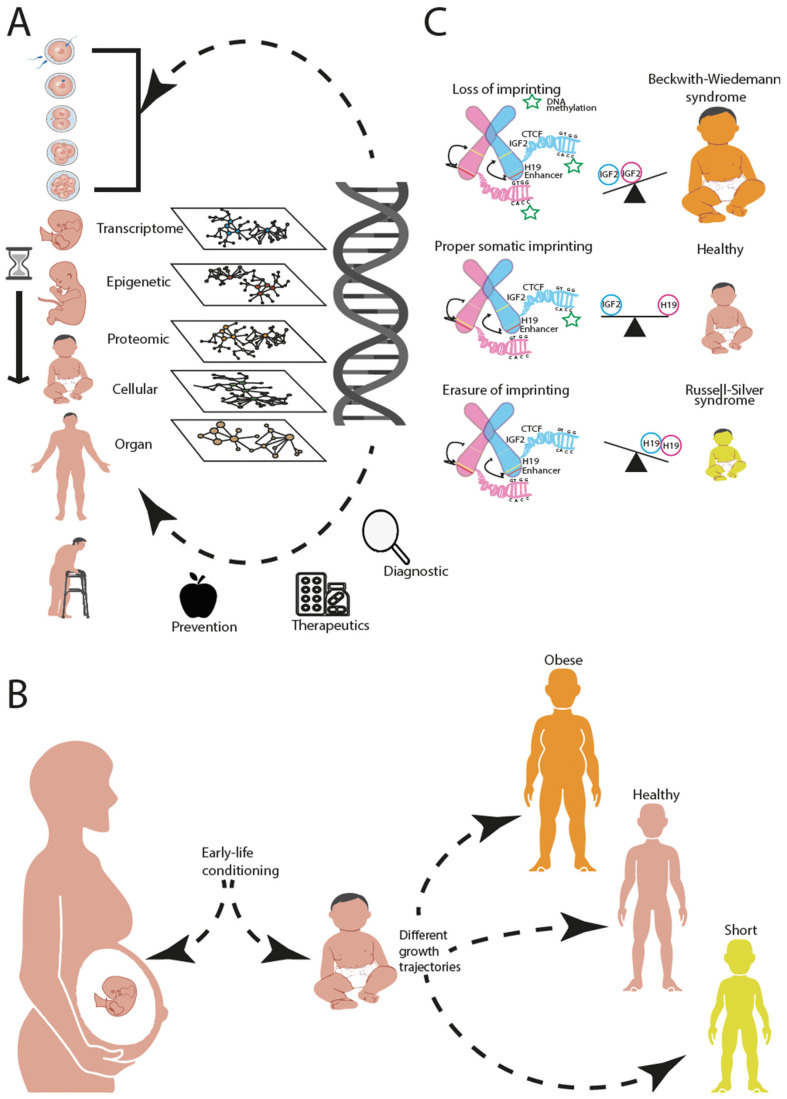
Dissecting biological complexity in the different layers of biological organisation helps in prevention, diagnosis and treatment. (**A**) Genetic perturbation during the periconceptional period (first two weeks after conception) can propagate through the different layers of biological networks: transcriptome, epigenome, proteome, cellular level and organ level leading to predisposition for disease phenotypes later on in life. Dissecting and integrating these biological layers are crucial for prevention, early diagnosis and potential treatments. (**B**) Early life conditioning can influence growth trajectories in life, contributing in predisposition for different phenotypes (health, short, and obese). (**C**) Epigenetic modifications, such as DNA methylation in particular regions of the DNA containing imprinting genes, could alter the normal genetic balance of the maternal and paternal alleles. As an example, we show the consequences of alterations of the IGF2-H19 imprinting gene balance, which can lead to either gigantism (Beckwith–Wiedemann syndrome) or nanism (Russell–Silver syndrome).

**Figure 2 genes-13-00764-f002:**
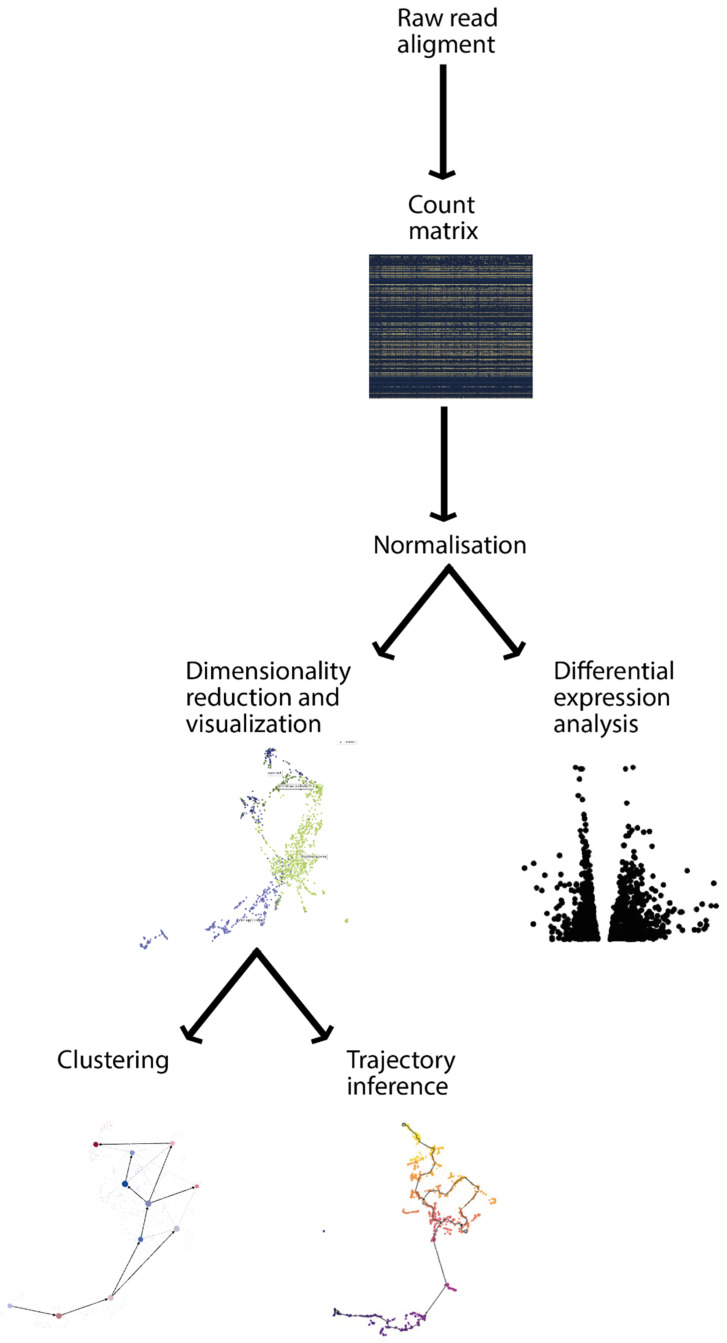
Transcriptomic analysis pipeline. Starting from the raw read alignments, several steps are needed to obtain concrete biological results, such as the identification of differential expressed genes or cluster marker genes. The first step is to align gene sequences to a gene annotation reference to be able to count the number of reads for each gene. This will allow us to obtain a count matrix, which can be used for differential expression analysis, identifying genes that are significantly changed (up/down regulated) in certain conditions and visualising them for example in a volcano plot. In parallel, the dimensions of the count matrix can be reduced and visualised with several techniques (i.e., PCA, t-SNE, and UMAP). This allows to identify clusters and, in the context of single-cell experiments, also to infer developmental trajectories.

**Figure 3 genes-13-00764-f003:**
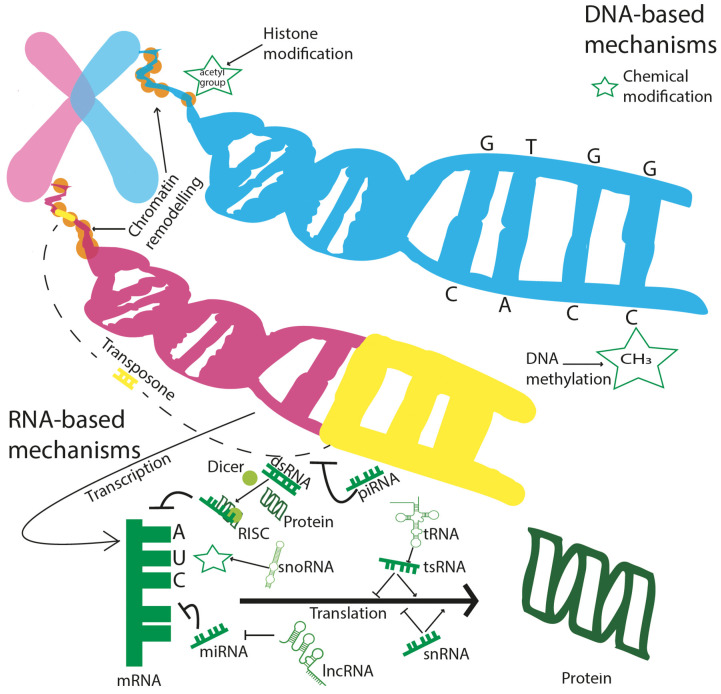
Epigenetic modifications occur on different biological scales. DNA-based mechanisms are concerned with histone modifications, consisting of chemical modifications (i.e., acetylation), DNA methylation, chromatin remodelling, and transposons. RNA-based mechanisms are multiple, complex, and still only partially known: miRNA and the RISC complex can induce mRNA degradation; lncRNAs silence the activity of miRNA, while snRNA and tsRNA can both silence, but also induce, mRNA translation to protein. piRNAs can interact with DNA, interfering with the genetic movements of transposons; snoRNAs induce chemical modifications at the mRNA level.

**Figure 4 genes-13-00764-f004:**
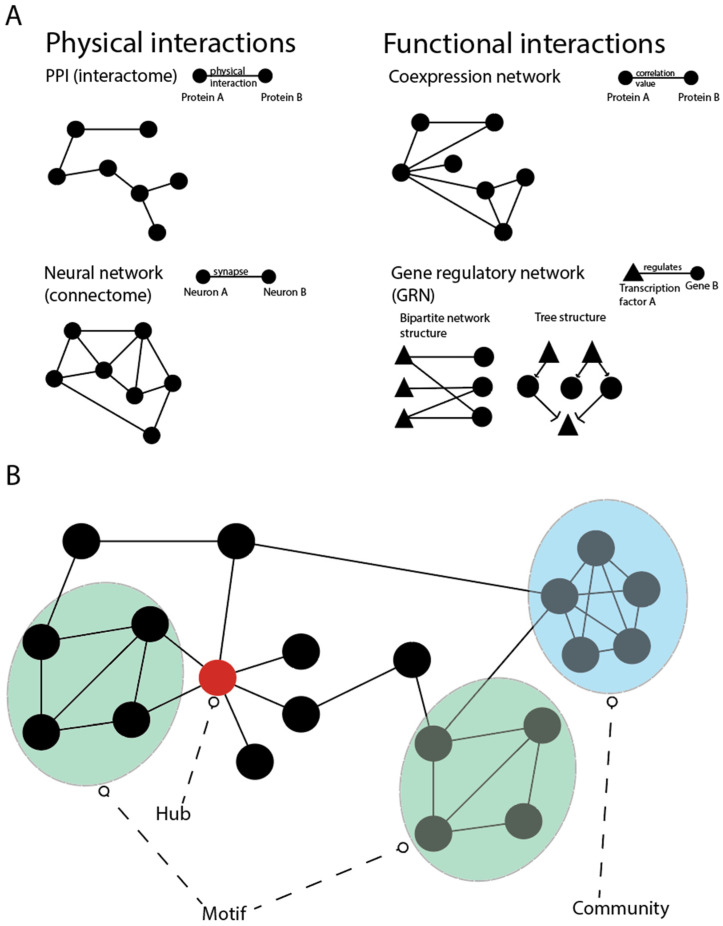
Biological networks and their topological characteristics. (**A**) Classification of biological networks in two major categories: physical and functional interactions. The first category includes the protein–protein interaction network (interactome), which represents the map of the physical interactions of all proteins and the neural network (connectome), which shows synapses that connects neurons. Networks that are constituted by functional interactions have edges that represents functional relationships, such as the level of expression (co-expression network) or the gene regulation (gene regulatory network). (**B**) The most important features of a network are hub (a node connected to many others), motif (recurrent structures in different parts of the network), and community (group of densely interconnected nodes).

**Table 1 genes-13-00764-t001:** Published transcriptomic datasets within the context of early human development.

Techniques	Sample Type	Number of Genes/Cells	Goals	Study
HG-U133 Plus 2.0 array (Affymetrix)	Oocytes	1361 transcripts expressed in oocytes	Study of oocyte transcriptomes	[220]
HG-U133 Plus 2.0 array (Affymetrix)	Oocytes	1514 overexpressed in oocytes compared with cumulus cells	Understanding of the mechanisms regulating oocyte maturation	[221]
HG-U133 Plus 2.0 array (Affymetrix)	Oocytes	5331 transcripts enriched in metaphase II oocytes relative to somatic cells	Comprehension of genes expressed in in vivo matured oocytes	[222]
HG-U133 Plus 2.0 array (Affymetrix)	Oocytes	10,183 genes were expressed in germinal vesicle	Study of global gene expression in human oocytes at the later stages of folliculogenesis (germinal vesicle stage)	[223]
HG-U133 Plus 2.0 array (Affymetrix)	Oocytes	Of the 8123 transcripts expressed in the oocytes, 374 genes showed significant differences in mRNA abundance in PCOS oocytes	Understanding of PCOS	[224]
HG-U133 Plus 2.0 array (Affymetrix)	Oocytes		Identification of new potential regulators and marker genes that are involved in oocyte maturation	[225]
HG-U133 Plus 2.0 array (Affymetrix)	Oocytes	283 genes found in the case report sample	Identification of molecular abnormalities in metaphase II (MII) oocytes	[226]
Whole Genome Bioarrays printed with 54,840 discovery probes representing 18,055 human genes and an additional 29,378 human expressed sequence tags (EST)	Oocytes	2000 genes were identified as expressed at more than 2-fold higher levels in oocytes matured in vitro than those matured in vivo	Analysis of the gene expression profile of oocytes following in vivo or in vitro maturation	[227]
Applied Biosystems Human Genome Survey Microarray (32,878 60-mer oligonucleotide)	Oocytes	Germinal vesicle, in vivo-MII and IVM-MII oocytes expressed 12,219, 9735 and 8510 genes, respectively	Characterisation of the patterns of gene expression in germinal vesicle stage and meiosis II oocytes matured in vitro or in vivo	[228]
HG-U133 Plus 2.0 array (Affymetrix)	Oocytes	342 genes showed a significantly different expression level between the two age groups (women aged 36 years (younger) and women aged 37–39 years (older))	Investigation of the effect of age on gene expression profile in mature oocytes	[229]
Two cDNA microarrays, each containing about 20,000 targets (representing in total ~29,778 independent genes according to Unigene Build 155)	Oocytes and embryos	1896 significant changes in expression following fertilization through day 3 of development	Global analysis of the preimplantation embryo transcriptome	[230]
cDNA microarrays containing 9600 cDNA spots	Oocytes and embryos	184, 29 and 65 genes were overexpressed in oocytes, 4- and 8-cell embryos, respectively	Identification of the differential expression profiles of genes in single oocytes, 4- and 8-cell preimplantation embryos	[231]
Genome Survey Microarrays V2.0 (Applied Biosystems)	Oocytes and embryos	107 DNA repair genes were detected in oocytes	Identification of the DNA repair pathways that may be active pre- and post-embryonic genome activation by investigating mRNA in human in vitro matured oocytes and blastocysts	[232]
HG-U133 Plus 2.0 array (Affymetrix)	Oocytes and embryos	5477 transcripts differentially expressed into transition from mature oocyte (MII) to 2-day embryo and 2989 transcripts differentially expressed into transition from 2-day to 3-day embryo	Study of global gene expression in human preimplantation development	[233]
HG-U133 Plus 2.0 array (Affymetrix)	Oocytes and embryos	45 eukaryotic initiation factors, 19 of which are differentially regulated between the 8-cell stage and blastocyst	Identification of gene networks behind cell fate decision in blastomeres	[234]
Illumina HiSeq2000 unpaired (TrueSeq)	Oocytes, embryos, and hESCs	124 single cells, 90 from 20 oocytes and embryos, 8 from primary hESC outgrowth, 26 from hESC passage 10, averaging 35.3 million reads per cell, average read length 100 bp. 22,687 maternally expressed genes detected, including 8701 lncRNAs, 2733 of them novel and developmental stage specific	Comparing the gene expression of human epiblast in vitro with hESCs	[235]
Illumina HiSeq2000 paired-end (TrueSeq)	Embryos	86 single cells	Validating known marker genes and highlighting differences between human and mouse pre-implantation development	[236]
Illumina HiSeq2000 single-end (Smart-seq2)	Embryos	1529 single cells from 88 embryos of various developmental stages, averaging 8500 expressed genes	Showcasing the differentiation of cell lineage in pre-implantation embryos and X-chromosome dosage compensation in females	[237]
Illumina HiSeq4000 paired-end (STRT-Seq and Trio-seq2)	Embryos	7636 single cells from 65 pre/post implantation embryos	Observation of genome regulation surrounding implantation	[238]

**Table 2 genes-13-00764-t002:** Resources containing epigenic data.

Name	Type of Data	URLs	Description	Reference
National Institutes of Health Roadmap Epigenome Project	DNA methylation Histone modifications Chromatin accessibility Small RNA-seq	www.roadmapepigenomics.org (accessed on 3 March 2022)	The consortium provides an analysis of stem cells and primary ex vivo tissues to collect normal epigenomes to provide a reference for comparison and integration in future studies.	[217]
ENCODE (Encyclopedia of DNA Elements Project)	DNA binding DNA accessibility DNA methylation Three-dimensional chromatin structure Replication timing Genotyping snATAC-seq DNA sequencing	https://www.encodeproject.org/ (accessed on 3 March 2022)	The consortium built a comprehensive parts list of functional elements in the human genome, including all the regulatory elements in different biological levels of complexity.	[218]
Human Epigenome Consortium	Histone modifications Chromatin accessibility Methylome Whole genome sequencing TF-binding sites	https://epigenomesportal.ca/ihec/ (accessed on 3 March 2022)	Large collection of studies containing human epigenome and transcriptome grouped by tissue and cell type.	[219]
Histone Infobase (HIstome)	Histone modifications	http://www.iiserpune.ac.in/~coee/histome/ (accessed on 3 March 2022)	Database covering 5 different types of histones, 8 types of their post-translational modification and 13 classes of modifying enzymes	[220]
DeepBlue	DNA methylation Histone modifications and variants DNasel Transcription factors binding sites Chromatin accessibility	https://deepblue.mpi-inf.mpg.de/ (accessed on 3 March 2022)	This source provides a great effort for integrating different databases and sources and obtaining a large comprehensive epigenomic consultable tool (via web interface or API interface)	[221]
MethBase	Methylome from different organisms	http://smithlabresearch.org/software/methbase/ (accessed on 3 March 2022)	For each methylome, they provide methylation level at individual sites, regions of allele specific methylation, hypo- or hyper-methylated regions, partially methylated regions, metadata and statistics.	[222]
iMETHYL	Methylome Whole genome sequencing	http://imethyl.iwate-megabank.org/ (accessed on 3 March 2022)	They provide a multi-omics data centering source for DNA methylation, also including information about cell types.	[223]
NONCODE	lncRNA	http://www.noncode.org/index.php (accessed on 3 March 2022)	This database comprises lncRNA from different organisms in health and disease.	[224]
miRBase	miRNA	https://www.mirbase.org/ (accessed on 3 March 2022)	This is a searchable database of published miRNA sequences and annotations.	[225]
PolymiRTS Database 3.0	miRNA	https://compbio.uthsc.edu/miRSNP/ (accessed on 3 March 2022)	Database containing miRNAs biological annotations, relationships with disease states and gene expression and their polymorphisms, variants and mutations.	[226]
snOPY	snoRNA	http://snoopy.med.miyazaki-u.ac.jp/ (accessed on 3 March 2022)	They provide a list of snoRNAs, snoRNA locus, target RNAs and orthologs for 39 different organisms.	[90]
snoDB	snoRNA	http://scottgroup.med.usherbrooke.ca/snoDB/ (accessed on 3 March 2022)	It harmonises human snoRNAs information from different sources, such as sequence databases and target information.	[91]
RMBase v2.0	RNA modification peaks and sites	http://rna.sysu.edu.cn/rmbase/ (accessed on 3 March 2022)	This database provides an important source for all the possible RNA modifications, including miRNA, snRNAs and snoRNAs.	[227]
mQTLdb	Methylome Genotype profiling	http://www.mqtldb.org/ (accessed on 3 March 2022)	They provide methylation and genotype data on mother–child pairs providing access to meQTL mapping across five different stages of life.	[228]
Methylomic trajectories across fetal brain development	Methylome	https://epigenetics.essex.ac.uk/fetalbrain/ (accessed on 3 March 2022)	DNA methylation across fetal brain development.	[229]
Methylation quantitative trait loci (mQTL) in the developing human brain and their enrichment in genomic regions associated with schizophrenia	Methylation quantitative trait loci	https://epigenetics.essex.ac.uk/mQTL/ (accessed on 3 March 2022)	DNA methylation quantitative trait loci of human fetal brain.	[230]

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
