# Peer review of "Network Approaches for Charting the Transcriptomic and Epigenetic Landscape of the Developmental Origins of Health and Disease"

_genes, 2022, doi:10.3390/genes13050764_

Round 1

Reviewer 1 Report

Lombardo et al review article provides a state of the art of RNA screening methods and literature for epigenetics analysis.

The article is well written and easy to follow, I don’t have any major comments regarding the review, although the article could mention machine learning in the vocabulary since both unsupervised (e.g., clustering which is mentioned in the article), and supervised methods are related to the analysis of transcriptomic dataset. This would broaden the article “searchability” reach to AI researchers.

Possible minor typo [REF] line 377

Reviewer 2 Report

Authors in this review manuscript provided a solid review of the major transcriptomic and epigenetic processes and respective datasets that are most relevant for studying the periconceptional period. Both basic data processing and more advanced data integration methods have been discussed. It is a timely review and easy to understand. However, I have the following concerns.

  1. Authors used four major sections: transcriptomics, epigenetics, networks, and integrative methods. Structure-wise, it is very easy to follow. However, these contents are not balances. Integrative methods are not as solid as three previous methods. It will be helpful to add more contents there.
  2. Authors focused quite a lot on data preprocessing, software choice, and analysis pipelines. However, how each layer can contribute to development is rarely discussed. It would be beneficial to the general audience to have better high-level biological insights, rather than just how to process each data type
  3. Why noncoding RNAs are in the epigenetic section, instead of the transcriptomic section?
  4. In the epigenetic section, histone modifications are playing important roles but are not adequately discussed in the current manuscript
  5. Table 2 title is very confusing. It is about available epigenetic databases, but consortia like ENCODE has many more functional genomic data other than epigenetic info. Also, in the table itself, RNA-seq has been listed as epigenetic, rather than transcriptomic

Minor questions:

  1. Figure 1 and the two tables can be reformatted to make them dense.

Round 2

Reviewer 2 Report

All my comments have been addressed